# A Network Pharmacology to Explore the Potential Targets of Canagliflozin and Dapagliflozin in Treating Atherosclerosis

**Jin Wang** [1,†], **Dongning Li** [2,†], **Weiwei Ju** [3,*] **and Hongli Wang** [4,*]

1    Division of Vascular Surgery, The Second Affiliated Hospital of Dalian Medical University, 467 Zhongshan Road, Dalian116000, China
2    Dalian Municipal Women and Children's Medical Center (Group), Dalian 116000, China
3    Division of Hepatobiliary and Pancreatic Surgery, Department of Surgery, The Second Hospital of Dalian Medical University, 467 Zhongshan Road, Dalian 116000, China
4    Department of Cardiology, The Second Hospital of Dalian Medical University, 467 Zhongshan Road, Dalian 116000, China
*    Correspondence: moneywei2016@163.com (W.J.); hongliwang@dmu.edu.cn (H.W.)
†    These authors contributed equally to this work.

**Abstract:** Background: Atherosclerosis (AS) is an important pathological basis of many cardiovascular diseases. Canagliflozin and dapagliflozin have yielded impressive results in the treatment of cardiovascular disease in both diabetic and non-diabetic patients. In this study, we investigated their targets and mechanism involved in the treatment of atherosclerosis using network pharmacology. Methods: The potential targets of canagliflozin and dapagliflozin were gathered from the database PharmMapper. Targets associated with AS were derived from the GeneCards, Drugbank, DisGeNet, and therapeutic target databases (TTD) by searching for keywords on atherosclerosis and coronary artery disease. Overlap targets were collected by uploading drug and disease targets into jvenn. The cross-targets of the Venny plots were uploaded to the STRING database, and a protein–protein interaction (PPI) was constructed with their calculated features, aiming to reveal several key targets. Key targets were selected by using a plug-in of the Cytoscape software. Gene ontology (GO) enrichment analysis and Kyoto Encyclopedia of Genes and Genomes (KEGG) pathway analysis were performed using the database Metascape. Cytoscape was used to set up the pathways-genes network. Molecular docking with core targets and drugs was performed with AutoDock. Results: A total of 288 canagliflozin targets, 287 dapagliflozin targets and 4939 AS-related targets were obtained. A total of 191 overlapping targets were found after intersecting. Five core targets, including protein kinase B (*Akt1*), Mitogen-activated protein kinase 1 (*MAPK1*), Mitogen-activated protein kinase 14 (*MAPK14*), Proto-oncogene tyrosine-protein kinase SRC (*SRC*) and Epidermal growth factor receptor (*EGFR*) were collected. Pathways, biological processes, molecular functions and cellular components of canagliflozin and dapagliflozin were found. Conclusion: Canagliflozin and dapagliflozin play a role in atherosclerosis by regulating *Akt1, MAPK1, MAPK14, SRC* and *EGFR*. Our research provides further insights into the use of canagliflozin and dapagliflozin in the treatment of atherosclerosis.

**Keywords:** network pharmacology; canagliflozin; dapagliflozin; potential targets; mechanism; atherosclerosis





## 1. Introduction

Atherosclerosis (AS) is a chronic inflammatory disease characterized by intense immune activity. Inflammation plays an important role in the occurrence and development of atherosclerotic lesions [1]. Atherosclerotic plaque formation is characterized by lipid accumulation, local inflammation, proliferation, apoptosis, necrosis and fibrosis of smooth muscle cells (SMCs) [2]. The main cause of atherosclerotic stenosis is the activation of inflammatory cells and a series of chronic inflammatory reactions after initial endothelial cell injury [3]. In the early stage of atherosclerosis, risk factors such as hyperlipidemia,

diabetes, smoking and hypertension can induce an oxidative stress response, activate cytokines and increase LDL levels. Meanwhile, macrophages migrate to the vascular wall, and inflammatory stimulators promote endothelial dysfunction and initiate atherosclerotic lesions [4]. At present, the most effective therapeutic drugs for AS are statins, which reduce the levels of atherogenic lipoproteins and prevent major cardiovascular events. However, treatment with statins is ineffective in reducing cholesterol levels in a small proportion of users, and prolonged use of statins may increase the risk of side effects [5]. Therefore, it is urgent to find new anti-AS drugs of satisfactory therapeutic effect and suitable for long-term use.

Canagliflozin and dapagliflozin, known as sodium-glucose cotransporter-2 (SGLT2) inhibitors, are a class of anti-diabetic compounds that lower blood glucose by selectively blocking the reabsorption of glucose in the proximal convoluted tubule (PCT) [6]. Recently, a growing number of studies have shown that canagliflozin and dapagliflozin are surprisingly effective in improving cardiovascular and kidney disease in both diabetic and non-diabetic patients. New clinical outcome trials have demonstrated that glucose-lowering drugs, especially SGLT2 inhibitors, overall reduce the risk of fatal and non-fatal atherosclerotic cardiovascular events and all-cause mortality [7]. Similarly, in two trials involving patients with type 2 diabetes and an elevated risk of cardiovascular disease, canagliflozin reduced the risk for primary cardiovascular outcome compared with those who received placebos [8]. In patients with heart failure and reduced ejection fraction, regardless of diabetes, dapagliflozin reduced the number of hospitalizations for heart failure or cardiovascular death compared to those who received placebos [9]. Some evidence from animal models indicates that SGLT2 inhibitors could prevent AS. Canagliflozin attenuated the progression of AS in HFD-fed ApoE$^{-/-}$ mice [10]. In addition, dapagliflozin has been shown to inhibit AS in ApoE$^{-/-}$ mice [11].

Network pharmacology is based on the concept of a multilevel and multiangle interaction network of disease-gene-target-drug [12]. It is associated with multiple disciplines in systems biology, pharmacology, computational biology, and network analysis [13,14]. Professional databases, resources, and software were used to visualize the relationship between drug–drug interaction and drug–disease interaction. Network pharmacology broke the traditional concept of single drugs, single target and single disease, and proposes that drugs act on multiple targets and multiple pathways [15]. Thus, network pharmacology has been an effective way to search for new research drugs and discover their potential mechanisms.

In this article, we used network pharmacology to demonstrate the underlying mechanisms of canagliflozin and dapagliflozin in the management of atherosclerosis. Therefore, using a variety of databases, we systematically predicted and analyzed the complex relationship between the disease, drugs and targets. Meanwhile, we discussed the potential mechanisms of some important targets of canagliflozin and dapagliflozin in the treatment of atherosclerosis. This study provides a scientific basis for further research on SGLT2 inhibitors for cardiovascular diseases. The process for our study can be found in Figure S1.

## 2. Materials and Methods

### 2.1. Prediction of Targets for Canagliflozin and Dapagliflozin

Two-dimensional structures of canagliflozin and dapagliflozin were searched in the PubChem (https://pubchem.ncbi.nlm.nih.gov/, accessed on 11 August 2021) database. The potential targets of canagliflozin and dapagliflozin were gathered from the database PharmMapper (http://www.lilab-ecust.cn/pharmmapper/, accessed on 26 August 2021). Names of target proteins were translated into gene names in the UniProt (http://www.uniprot.org/, accessed on 26 August 2021) database [16]. Potential targets were deleted if their names were not found in the Uniprot database.

## 2.2. Collection of Disease Targets of Atherosclerosis

The AS-related genes were derived from the GeneCards (https://www.GeneCards.org/, accessed on 11 August 2021) [17], Drugbank [18], DisGeNet [19] and TTD [20] databases by searching with the key words of Atherosclerosis and Coronary artery disease. Duplicate genes were deleted after putting all target genes together.

## 2.3. Venn Diagram Plotting

The website of jvenn (http://jvenn.toulouse.inra.fr/app/index.html, accessed on 26 August 2021) was used to plot Venn diagram by uploading both drug targets and disease targets. Intersecting genes were the potential targets of canagliflozin and dapagliflozin, and they overlapped with Atherosclerosis [21].

## 2.4. Protein-Protein Interaction (PPI)

Intersecting genes from Venny plots were uploaded to the STRING database (https://string-db.org/, accessed on 27 August 2021), and Homo sapiens was selected for species [22]. The protein–protein interaction (PPI) networks of 191 genes were created with a high confidence level (interaction score > 0.700), and irrelevant targets were concealed. The result from STRING database was downloaded and imported into Cytoscape3.7.2 U.S. (https://cytoscape.org/, accessed on 27 August 2021) [23], then the topology parameters (degrees) of the nodes were calculated, and core targets were defined based on the degree. The potential targets of canagliflozin and dapagliflozin were imported into Cytoscape3.7.2 software to construct networks.

## 2.5. Gene Functions and Pathway Enrichment Analysis with Potential Targets

Gene ontology (GO) enrichment analysis and Kyoto Encyclopedia of Genes and Genomes (KEGG) pathway analysis were carried out using the database Metascape (https://metascape.org/gp/index.html#/main/step1, accessed on 30 August 2021) [24]. The top 20 results from the GO enrichment and KEGG pathway enrichment analyses were presented. GO enrichment analysis mainly covers three aspects of biology, namely biological process, molecular function and cellular component. It is widely used in the field of gene function classification and function distribution prediction of targets.

## 2.6. Construction of Target Gene-Drug Network

Target genes and the top 20 results of KEGG pathway enrichment analyses were analyzed by Cytoscape3.7.2 software for visual network analysis. In the network, nodes represented genes and pathways, and edges represented interactions between the nodes [21].

The importance of the gene or pathway is assessed by the topology parameter (degree). The degree of a node is the number of edges connected to the node, and the greater the degree, the more important the node is in the network.

## 2.7. Molecular Docking of the Target Gene

The rigid docking analysis was performed by AutoDock 4.2 with MGL tools 1.5.6 (The Scripps Research Institutes, San Diego, CA, USA). The PDB files of canagliflozin and dapagliflozin were produced using Chem3D Pro software. The crystal structures of AKT1 (PDBID: 6HHG), MAPK1 (PDBID: 4XJ0), MAPK14 (PDBID: 4L8M), RHOA (PDBID: 1A2B), SRC (PDBID: 2H8H) and EGFR (PDBID: 5UG9) were downloaded from RCSB Protein Data Bank. Following the requirement of the docking study, ions, water molecules and non-standard amino acid residues were removed from the proteins. For the docking case, the model with the lowest energy was selected as the binding mode for analysis. The output from AutoDock was rendered by the PyMol program [25].

## 3. Results

### 3.1. Network Construction of Drugs and Targets

A total of 288 canagliflozin targets and 287 dapagliflozin targets were identified by using the PharmMapper database. Target names were translated into gene names from the UniProt database. Details of these targets are listed in Tables S1 and S2, and the maps of the drug-target networks are shown in Figure 1. The hexagons represent canagliflozin and dapagliflozin, and the squares represent the targets.

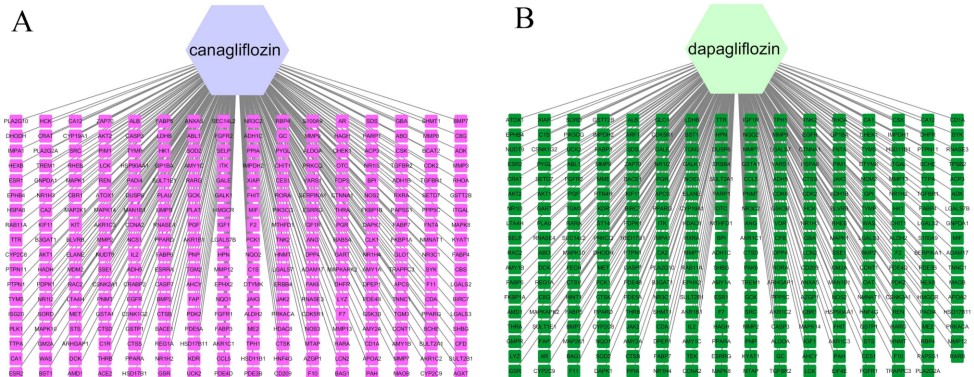

**Figure 1.** The targets of canagliflozin (**A**) and dapagliflozin (**B**). Note: hexagons represent drugs, squares represent targets.

### 3.2. Targets of Atherosclerosis

By searching for Atherosclerosis and Coronary artery disease in the database, 92 targets were found in the Drugbank database, 1061 targets were found in the DisGeNet database after screening for scores above the median, 4708 targets were found in the GeneCards database after screening for scores above the median, and 26 targets were found in the TTD database. A total of 4939 AS-related targets were found after the duplicates were deleted.

### 3.3. Prediction of Canagliflozin and Dapagliflozin Targets in Atherosclerosis

After uploading targets of drugs and disease to jvenn, 191 overlapping targets were found, as shown in Figure 2. These targets might be the key genes of canagliflozin and dapagliflozin in the treatment of atherosclerosis.

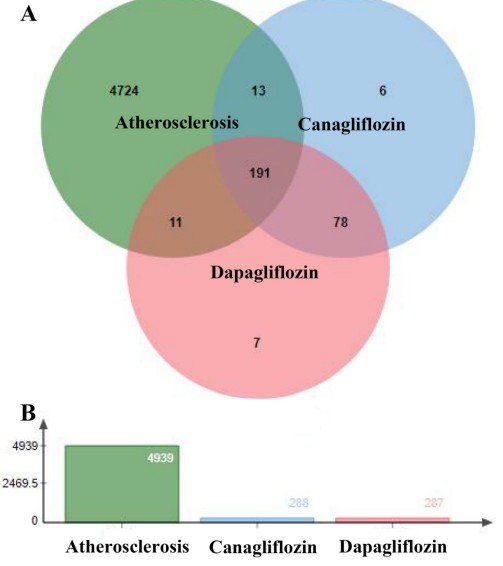

**Figure 2.** Intersection targets of drugs and atherosclerosis. (**A**) Venn diagram of drug targets and atherosclerosis targets. (**B**) Histograms of atherosclerosis targets and drug targets.

### 3.4. Construction of PPI Networks

In order to discover the possible mechanisms of canagliflozin and dapagliflozin in treating atherosclerosis, the STRING database was used to construct the PPI network of 191 targets shared by drugs and disease as shown in Figure 3A. The information derived from the STRING database was imported into the Cytoscape3.7.2 software for further analysis, and a visualized PPI network was constructed, with the dot size and color reflecting the degree of freedom. The higher the degree of freedom, the more biological functions were involved. The intensity increased from outside to inside, which was 1–1, 2–3, 4–6, 7–11, 12–16, 17–46, respectively. The results are shown in Figure 3B. Then, three algorithms (degree of freedom, maximum neighborhood component (MNC), and maximal clique centrality (MCC)) in CytoHubba plug-in were used to analyze each node in the PPI network, and six genes (*RHOA*, *AKT1*, *EGFR*, *MAPK1*, *MAPK14*, and *SRC*) were screened out by taking the intersection of three results. Their network of interactions are shown in Figure 3C,D.

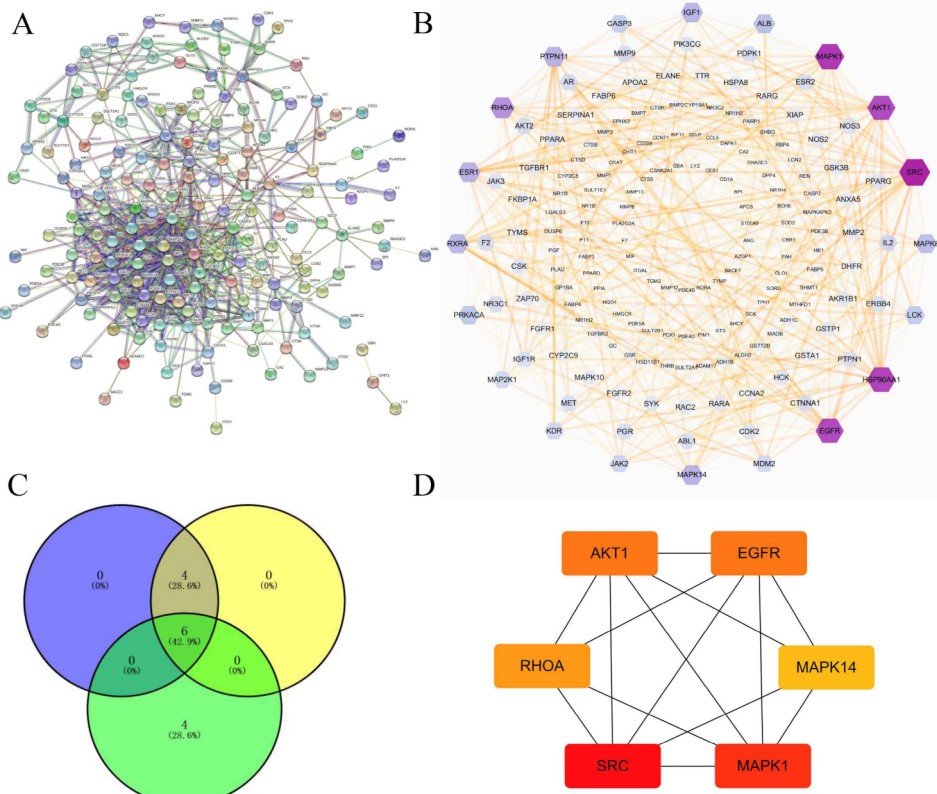

**Figure 3.** The protein–protein interaction (PPI) network of intersection targets of drugs and disease. (**A**) The PPI network of 191 intersection targets for canagliflozin and dapagliflozin in treating atherosclerosis. Note: In the network diagram, the nodes represent each protein, and the node label is the name of the represented protein. The pattern in the node represents the three-dimensional structure of the protein. If it is empty, the structure is currently unknown. If there is an interaction between the two proteins, it is connected by a connecting line. The color of the line reflects the type of interaction, including experimentally verified or predicted, and also includes direct physical interaction, co-expressed gene fusion and other relationships. (**B**) The PPI network showing detailed interactions of the targets constructed by Cytoscape. Note: The result from STRING database was downloaded and imported into Cytoscape, then the topology parameters (degrees) of the nodes were calculated, and core targets were defined based on the degree. The larger the volume of the hexagon and the darker the color, the more important the target is. (**C**) The top 10 genes were calculated from the PPI network by the degree of freedom, MNC, and MCC, and the overlapping genes were then screened by Venn diagrams. (**D**) The PPI network of six overlapping genes.

*3.5. GO Enrichment Analysis and KEGG Pathway Enrichment Analysis*

To explore the underlying drug mechanism in managing AS, 191 predicted targets were entered into the Metascape database for GO enrichment analysis and KEGG pathway analysis.

### 3.5.1. KEGG Pathway Enrichment Analysis

The 20 main pathways were found following the KEGG pathway enrichment analysis (Figure 4 and Supplementary Figure S2). They were mainly involved in pathways in cancer (hsa05200), proteoglycans in cancer (hsa05205), endocrine resistance (hsa01522), fluid shear stress and atherosclerosis (hsa05418), Epstein–Barr virus infection (hsa05169), th17 cell differentiation (hsa04659), transcriptional misregulation in cancer (hsa05202), adherens junction (hsa04520), microRNAs in cancer (hsa05206), PPAR signaling pathway (hsa03320), IL-17 signaling pathway (hsa04657), epithelial cell signaling in helicobacter pylori infection (hsa05120), viral carcinogenesis (hsa05203), platinum drug resistance (hsa01524), chemical carcinogenesis (hsa05204), longevity regulating pathway (hsa04211), natural killer cell-mediated cytotoxicity (hsa04650), arachidonic acid metabolism (hsa00590), complement and coagulation cascades (hsa04610), and apoptosis-multiple species (hsa04215). Details on these pathways are listed in Table 1.

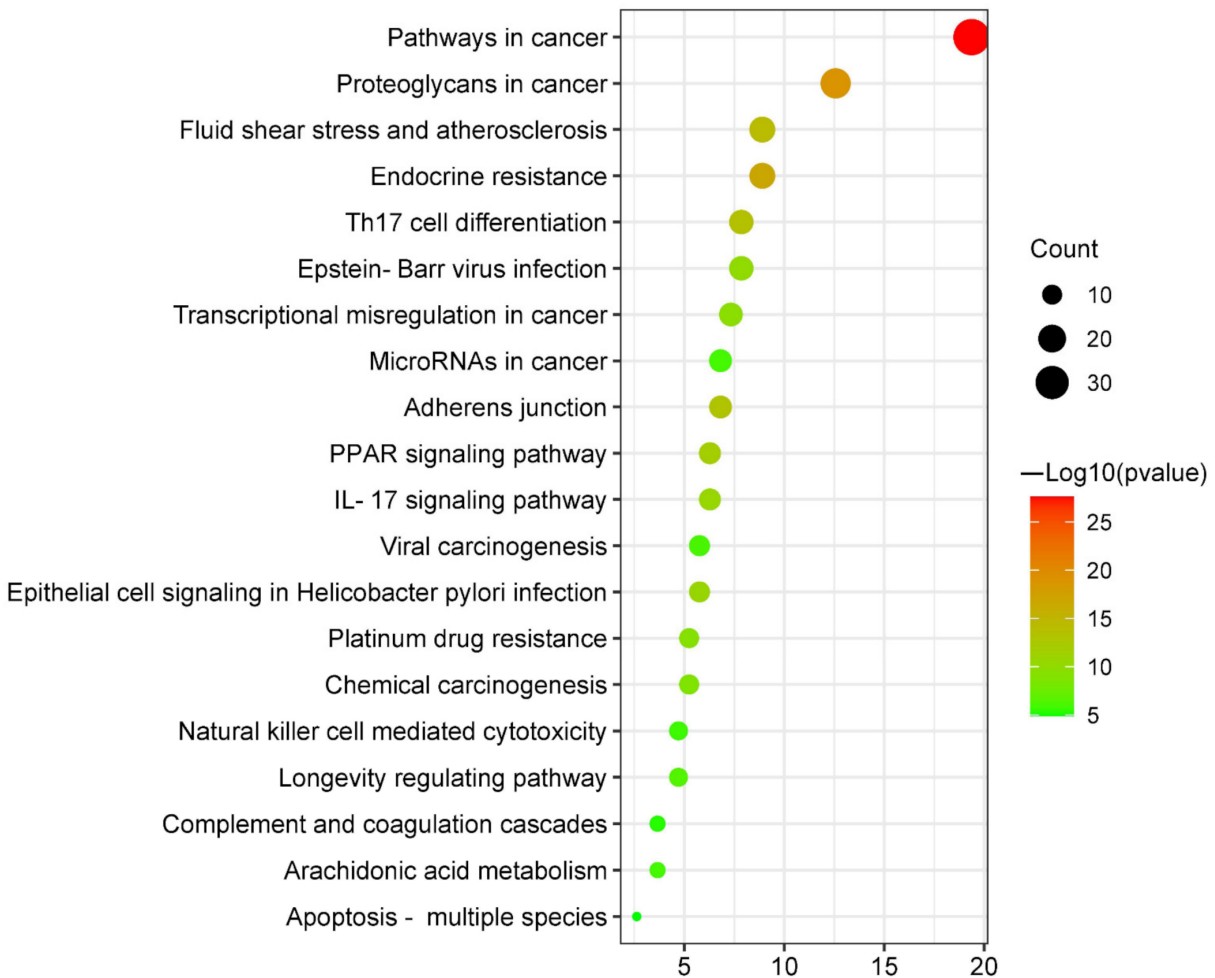

**Figure 4.** Bubble plot of KEGG pathway analysis of targets shared by canagliflozin, dapagliflozin and atherosclerosis.

**Table 1.** KEGG pathway analysis of targets shared by canagliflozin, dapagliflozin and Atherosclerosis.

| GO | Description | Gene Ratio | $p$ | Count |
|---|---|---|---|---|
| hsa05200 | Pathways in cancer | 19.37 | $2.45471 \times 10^{-28}$ | 37 |
| hsa05205 | Proteoglycans in cancer | 12.57 | $1.51356 \times 10^{-19}$ | 24 |
| hsa01522 | Endocrine resistance | 8.9 | $1.44544 \times 10^{-17}$ | 17 |
| hsa05418 | Fluid shear stress and atherosclerosis | 8.9 | $6.76083 \times 10^{-15}$ | 17 |
| hsa04659 | Th17 cell differentiation | 7.85 | $3.23594 \times 10^{-14}$ | 15 |
| hsa04520 | Adherens junction | 6.81 | $7.58578 \times 10^{-14}$ | 13 |
| hsa03320 | PPAR signaling pathway | 6.28 | $1.58489 \times 10^{-12}$ | 12 |
| hsa05120 | Epithelial cell signaling in Helicobacter pylori infection | 5.76 | $1.86209 \times 10^{-11}$ | 11 |
| hsa04657 | IL-17 signaling pathway | 6.28 | $2.63027 \times 10^{-11}$ | 12 |
| hsa05169 | Epstein–Barr virus infection | 7.85 | $1.09648 \times 10^{-10}$ | 15 |
| hsa05202 | Transcriptional misregulation in cancer | 7.33 | $2.81838 \times 10^{-10}$ | 14 |
| hsa01524 | Platinum drug resistance | 5.24 | $6.60693 \times 10^{-10}$ | 10 |
| hsa05204 | Chemical carcinogenesis | 5.24 | $1.90546 \times 10^{-9}$ | 10 |
| hsa04211 | Longevity regulating pathway | 4.71 | $3.63078 \times 10^{-7}$ | 9 |
| hsa05203 | Viral carcinogenesis | 5.76 | $8.12831 \times 10^{-7}$ | 11 |
| hsa05206 | MicroRNAs in cancer | 6.81 | $1.14815 \times 10^{-6}$ | 13 |
| hsa00590 | Arachidonic acid metabolism | 3.66 | $1.14815 \times 10^{-6}$ | 7 |
| hsa04650 | Natural killer cell-mediated cytotoxicity | 4.71 | $1.7378 \times 10^{-6}$ | 9 |
| hsa04610 | Complement and coagulation cascades | 3.66 | $5.49541 \times 10^{-6}$ | 7 |
| hsa04215 | Apoptosis-multiple species | 2.62 | $1.25893 \times 10^{-5}$ | 5 |

### 3.5.2. Biological Process Enrichment Analysis

After the biological process enrichment analysis in the GO enrichment analysis, 20 results were selected based on their $p$ values and number of enrichments (Figure 5 and Supplementary Figure S3). The predicted targets were included in cellular response to hormone stimulus (GO:0032870), cellular response to lipid (GO:0071396), muscle cell proliferation (GO:0033002), response to lipopolysaccharide (GO:0032496), wound healing (GO:0042060), positive regulation of cell migration (GO:0030335), organic hydroxy compound metabolic process (GO:1901615), regulation of kinase activity (GO:0043549), regulation of inflammatory response (GO:0050727), response to nutrient levels (GO:0031667), cellular response to chemical stress (GO:0062197), epithelial cell differentiation (GO:0030855), regulation of cell activation (GO:0050865), positive regulation of cell death (GO:0010942), leukocyte migration (GO:0050900), response to decreased oxygen levels (GO:0036293), response to drug (GO:0042493), regulation of establishment of protein localization (GO:0070201), reactive oxygen species metabolic process (GO:0072593), and negative regulation of intracellular signal transduction (GO:1902532). Details about these biological processes are listed in Table 2.

**Table 2.** Biological process enrichment analysis of targets shared by canagliflozin, dapagliflozin and atherosclerosis.

| GO | Description | Gene Ratio | $p$ | Count |
|---|---|---|---|---|
| GO:0032870 | cellular response to hormone stimulus | 20.94 | $3.71535 \times 10^{-24}$ | 40 |
| GO:0071396 | cellular response to lipid | 19.9 | $1.38038 \times 10^{-22}$ | 38 |
| GO:0033002 | muscle cell proliferation | 14.66 | $2.75423 \times 10^{-22}$ | 28 |
| GO:0032496 | response to lipopolysaccharide | 15.71 | $5.88844 \times 10^{-21}$ | 30 |
| GO:0042060 | wound healing | 16.75 | $1.07152 \times 10^{-20}$ | 32 |
| GO:0030335 | positive regulation of cell migration | 18.32 | $1.38038 \times 10^{-20}$ | 35 |
| GO:1901615 | organic hydroxy compound metabolic process | 17.8 | $3.89045 \times 10^{-20}$ | 34 |
| GO:0043549 | regulation of kinase activity | 19.9 | $3.71535 \times 10^{-19}$ | 38 |
| GO:0050727 | regulation of inflammatory response | 15.18 | $7.24436 \times 10^{-19}$ | 29 |
| GO:0031667 | response to nutrient levels | 16.23 | $1.02329 \times 10^{-18}$ | 31 |

**Table 2.** *Cont.*

| GO | Description | Gene Ratio | $p$ | Count |
|---|---|---|---|---|
| GO:0062197 | cellular response to chemical stress | 14.14 | $3.01995 \times 10^{-18}$ | 27 |
| GO:0030855 | epithelial cell differentiation | 17.28 | $5.49541 \times 10^{-17}$ | 33 |
| GO:0050865 | regulation of cell activation | 17.28 | $7.24436 \times 10^{-17}$ | 33 |
| GO:0010942 | positive regulation of cell death | 16.75 | $1.38038 \times 10^{-16}$ | 32 |
| GO:0050900 | leukocyte migration | 13.61 | $2.51189 \times 10^{-16}$ | 26 |
| GO:0036293 | response to decreased oxygen levels | 12.57 | $1.28825 \times 10^{-15}$ | 24 |
| GO:0042493 | response to drug | 13.61 | $1.28825 \times 10^{-15}$ | 26 |
| GO:0070201 | regulation of establishment of protein localization | 15.18 | $2.0893 \times 10^{-15}$ | 29 |
| GO:0072593 | reactive oxygen species metabolic process | 10.99 | $5.24807 \times 10^{-15}$ | 21 |
| GO:1902532 | negative regulation of intracellular signal transduction | 14.66 | $1.1749 \times 10^{-14}$ | 28 |

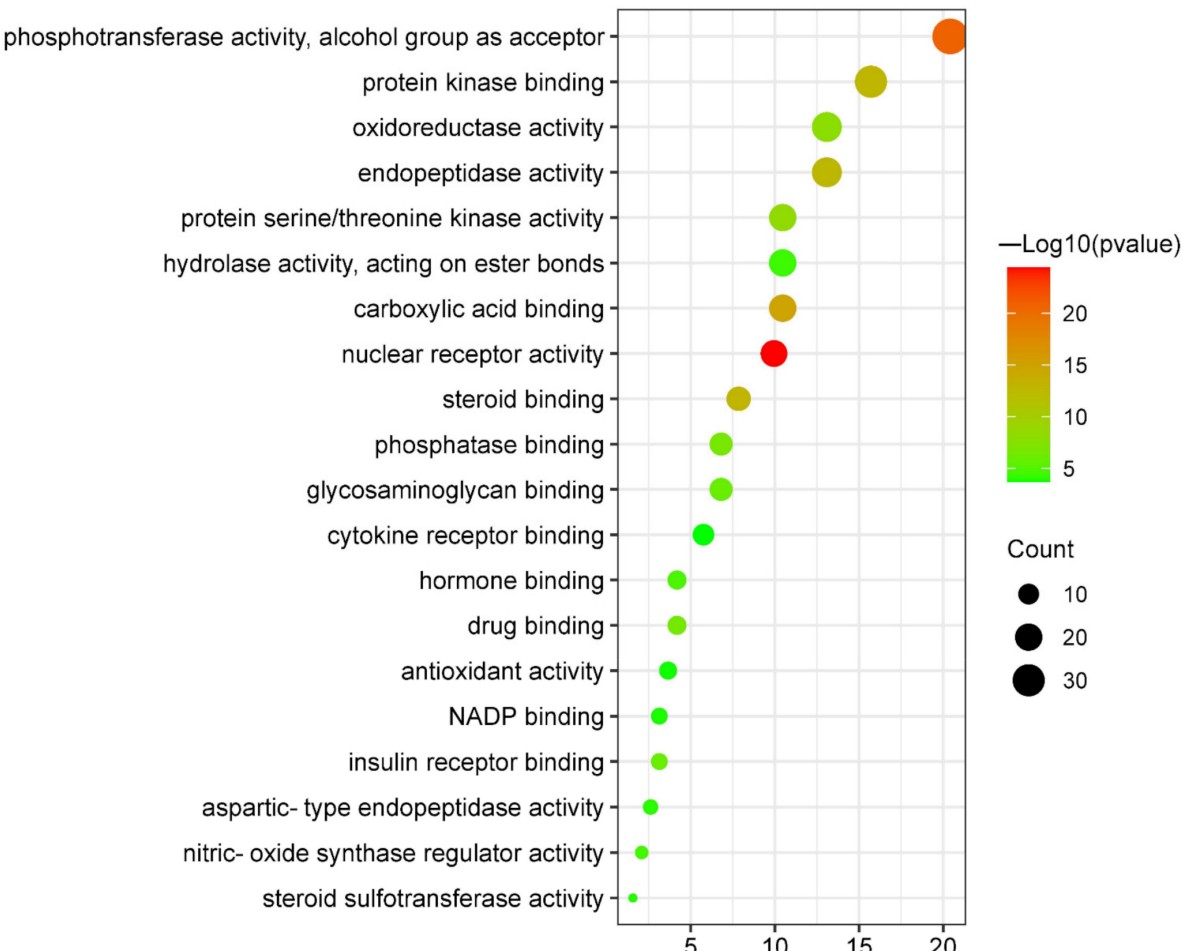

**Figure 5.** Bubble plot of biological process enrichment analysis of targets shared by canagliflozin, dapagliflozin and atherosclerosis.

3.5.3. Molecular Functions Enrichment Analysis

Canagliflozin and dapagliflozin might affect the following molecular functions (Figure 6 and Supplementary Figure S4) to improve AS: nuclear receptor activity (GO:0004879), phosphotransferase activity, alcohol group as acceptor (GO:0016773), carboxylic acid binding (GO:0031406), steroid binding (GO:0005496), protein kinase binding (GO:0019901), endopeptidase activity (GO:0004175), protein serine/threonine kinase activity (GO:0004674), oxidoreductase activity (GO:0016491), phosphatase binding (GO:0019902), drug binding (GO:0008144), glycosaminoglycan binding (GO:0005539), insulin receptor binding

(GO:0005158), hormone binding (GO:0042562), nitric-oxide synthase regulator activity (GO:0030235), hydrolase activity, acting on ester bonds (GO:0016788), aspartic-type endopeptidase activity (GO:0004190), steroid sulfotransferase activity (GO:0050294), NADP binding (GO:0050661), antioxidant activity (GO:0016209), and cytokine receptor binding (GO:0005126). Details about these molecular functions are listed in Table 3.

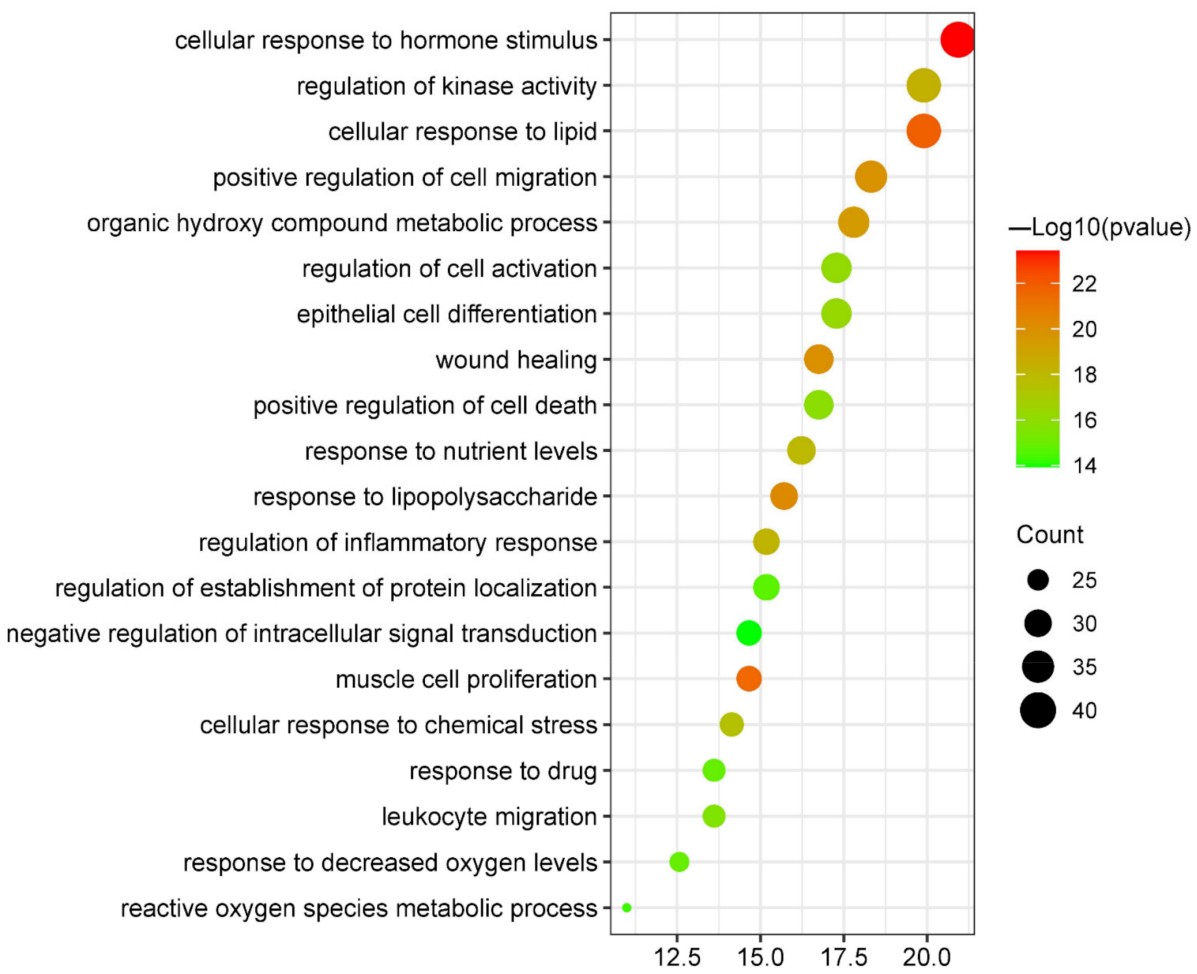

**Figure 6.** Bubble plot of molecular functions enrichment analysis of targets shared by canagliflozin, dapagliflozin and atherosclerosis.

**Table 3.** Molecular functions enrichment analysis of targets shared by canagliflozin, dapagliflozin and Atherosclerosis.

| GO | Description | Gene Ratio | *p* | Count |
|----|-------------|-----------|-----|-------|
| GO:0004879 | nuclear receptor activity | 9.95 | $3.80 \times 10^{-25}$ | 19 |
| GO:0016773 | phosphotransferase activity, alcohol group as acceptor | 20.42 | $1.45 \times 10^{-21}$ | 39 |
| GO:0031406 | carboxylic acid binding | 10.47 | $1.41 \times 10^{-15}$ | 20 |
| GO:0005496 | steroid binding | 7.85 | $1.26 \times 10^{-13}$ | 15 |
| GO:0019901 | protein kinase binding | 15.71 | $1.58 \times 10^{-13}$ | 30 |
| GO:0004175 | endopeptidase activity | 13.09 | $2.00 \times 10^{-13}$ | 25 |
| GO:0004674 | protein serine/threonine kinase activity | 10.47 | $4.47 \times 10^{-9}$ | 20 |
| GO:0016491 | oxidoreductase activity | 13.09 | $1.02 \times 10^{-8}$ | 25 |
| GO:0019902 | phosphatase binding | 6.81 | $1.91 \times 10^{-7}$ | 13 |
| GO:0008144 | drug binding | 4.19 | $2.57 \times 10^{-7}$ | 8 |
| GO:0005539 | glycosaminoglycan binding | 6.81 | $1.17 \times 10^{-6}$ | 13 |

**Table 3.** *Cont.*

| GO | Description | Gene Ratio | *p* | Count |
|---|---|---|---|---|
| GO:0005158 | insulin receptor binding | 3.14 | $1.26 \times 10^{-6}$ | 6 |
| GO:0042562 | hormone binding | 4.19 | $1.48 \times 10^{-5}$ | 8 |
| GO:0030235 | nitric-oxide synthase regulator activity | 2.09 | $2.63 \times 10^{-5}$ | 4 |
| GO:0016788 | hydrolase activity, acting on ester bonds | 10.47 | $3.80 \times 10^{-5}$ | 20 |
| GO:0004190 | aspartic-type endopeptidase activity | 2.62 | $9.12 \times 10^{-5}$ | 5 |
| GO:0050294 | steroid sulfotransferase activity | 1.57 | $1.05 \times 10^{-4}$ | 3 |
| GO:0050661 | NADP binding | 3.14 | $1.35 \times 10^{-4}$ | 6 |
| GO:0016209 | antioxidant activity | 3.66 | $1.78 \times 10^{-4}$ | 7 |
| GO:0005126 | cytokine receptor binding | 5.76 | $2.14 \times 10^{-4}$ | 11 |

### 3.5.4. Cellular Components Enrichment Analysis

After cellular components enrichment analysis, the top 20 results were collected (Figure 7 and Supplementary Figure S5). They were primarily involved in vesicle lumen (GO:0031983), membrane raft (GO:0045121), extracellular matrix (GO:0031012), ficolin-1-rich granule (GO:0101002), focal adhesion (GO:0005925), side of membrane (GO:0098552), receptor complex (GO:0043235), tertiary granule lumen (GO:1904724), endoplasmic reticulum lumen (GO:0005788), endolysosome lumen (GO:0036021), extrinsic component of cytoplasmic side of plasma membrane (GO:0031234), postsynapse (GO:0098794), protein kinase complex (GO:1902911), endocytic vesicle (GO:0030139), late endosome (GO:0005770), calcium channel complex (GO:0034704), dendrite (GO:0030425), perinuclear region of cytoplasm (GO:0048471), actin cytoskeleton (GO:0015629), and platelet alpha granule (GO:0031091). Details about these cellular components are listed in Table 4.

**Table 4.** Cellular components enrichment analysis of targets shared by canagliflozin, dapagliflozin and atherosclerosis.

| GO | Description | Gene Ratio | *p* | Count |
|---|---|---|---|---|
| GO:0031983 | vesicle lumen | 12.04 | $1.32 \times 10^{-13}$ | 23 |
| GO:0045121 | membrane raft | 11.52 | $7.76 \times 10^{-13}$ | 22 |
| GO:0031012 | extracellular matrix | 12.57 | $2.95 \times 10^{-10}$ | 24 |
| GO:0101002 | ficolin-1-rich granule | 7.85 | $5.75 \times 10^{-10}$ | 15 |
| GO:0005925 | focal adhesion | 9.42 | $1.12 \times 10^{-7}$ | 18 |
| GO:0098552 | side of membrane | 10.99 | $2.69 \times 10^{-7}$ | 21 |
| GO:0043235 | receptor complex | 9.42 | $2.82 \times 10^{-6}$ | 18 |
| GO:1904724 | tertiary granule lumen | 3.66 | $9.55 \times 10^{-6}$ | 7 |
| GO:0005788 | endoplasmic reticulum lumen | 6.28 | $1.35 \times 10^{-4}$ | 12 |
| GO:0036021 | endolysosome lumen | 1.57 | $2.14 \times 10^{-4}$ | 3 |
| GO:0031234 | extrinsic component of cytoplasmic side of plasma membrane | 3.66 | $4.07 \times 10^{-4}$ | 7 |
| GO:0098794 | postsynapse | 6.81 | $1.95 \times 10^{-2}$ | 13 |
| GO:1902911 | protein kinase complex | 2.62 | $2.19 \times 10^{-2}$ | 5 |
| GO:0030139 | endocytic vesicle | 4.71 | $2.24 \times 10^{-2}$ | 9 |
| GO:0005770 | late endosome | 4.19 | $3.09 \times 10^{-2}$ | 8 |
| GO:0034704 | calcium channel complex | 2.09 | $3.89 \times 10^{-2}$ | 4 |
| GO:0030425 | dendrite | 6.28 | $5.50 \times 10^{-2}$ | 12 |
| GO:0048471 | perinuclear region of cytoplasm | 6.81 | $6.03 \times 10^{-2}$ | 13 |
| GO:0015629 | actin cytoskeleton | 5.24 | $8.71 \times 10^{-2}$ | 10 |
| GO:0031091 | platelet alpha granule | 2.09 | $1.07 \times 10^{-1}$ | 4 |

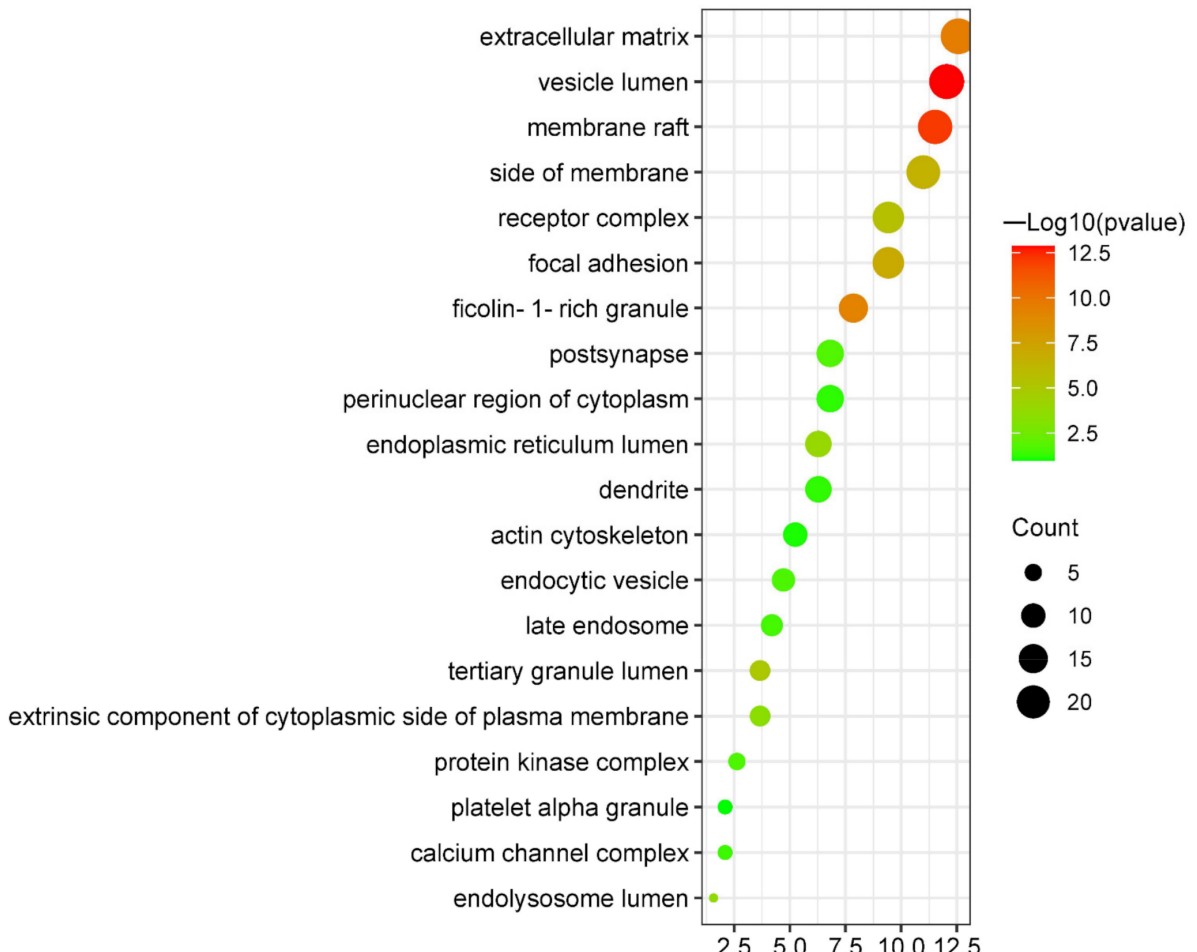

**Figure 7.** Bubble plot of cellular components enrichment analysis of targets shared by canagliflozin, dapagliflozin and atherosclerosis.

### 3.6. Network Construction of Targets-Pathways

The relationship between KEGG pathways and pathway-related genes is shown in Figure 8. There were 125 nodes and 270 edges in this network. The hexagon represented the genes, and V represented the pathways. These results further suggested that *Akt1*, *AKT2*, *MMP9*, *MDM2*, *CASP3*, *MAPK1*, *MAPK10*, *MAPK8*, *MAPK14*, *RHOA*, *SRC*, *MET*, and *EGFR* were key targets for canagliflozin and dapagliflozin to improve AS. In addition, *Akt1*, *MAPK1*, *MAPK14*, *RHOA*, *SRC* and *EGFR* were six hub targets in the PPI network of 191 targets shared by drugs and disease.

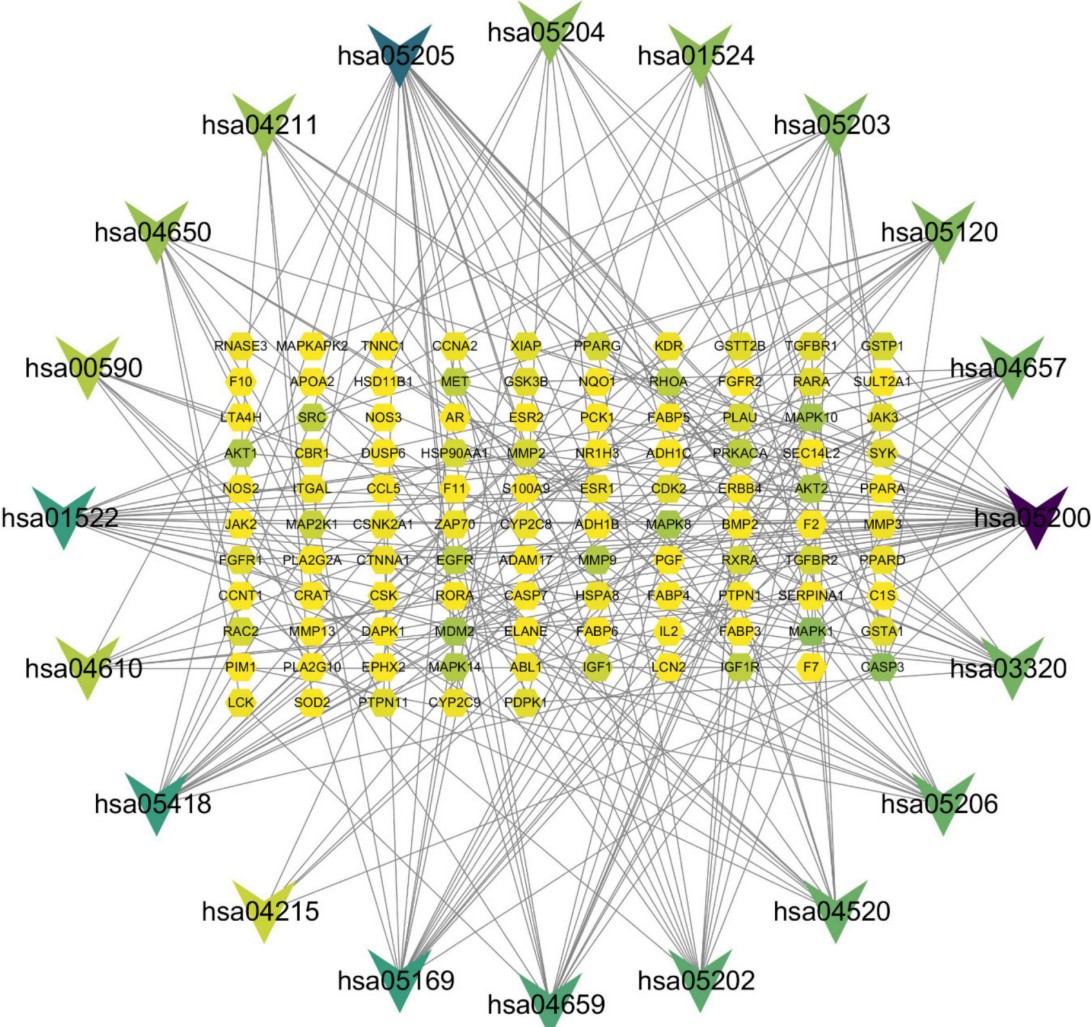

**Figure 8.** The network of pathways and pathway-related targets. Note: Hexagons represent the 105 targets, inverted triangles represent the top 20 pathways. The color of a node represents its degree, and the darker the color, the more important the node.

*3.7. Molecular Docking*

Canagliflozin and dapagliflozin were molecularly docked with six potential targets including Akt1 (PDBID: 6HHG), MAPK1 (PDBID: 4XJ0), MAPK14 (PDBID: 4L8M), RHOA (PDBID: 1A2B), SRC (PDBID: 2H8H) and EGFR (PDBID: 5UG9). A total of 12 pairs of receptor–ligand combinations were obtained (Figure 9), and details about these combinations are listed in Tables 5 and 6. The bindings in RHOA-canagliflozin (−5.59 kcal/mol) and RHOA-dapagliflozin (−6.90 kcal/mol) were weak and thus cannot be considered core targets. The average value of the other 10 combinations was −8.897 kcal/mol, suggesting that the binding between the core targets and the drugs was strong.

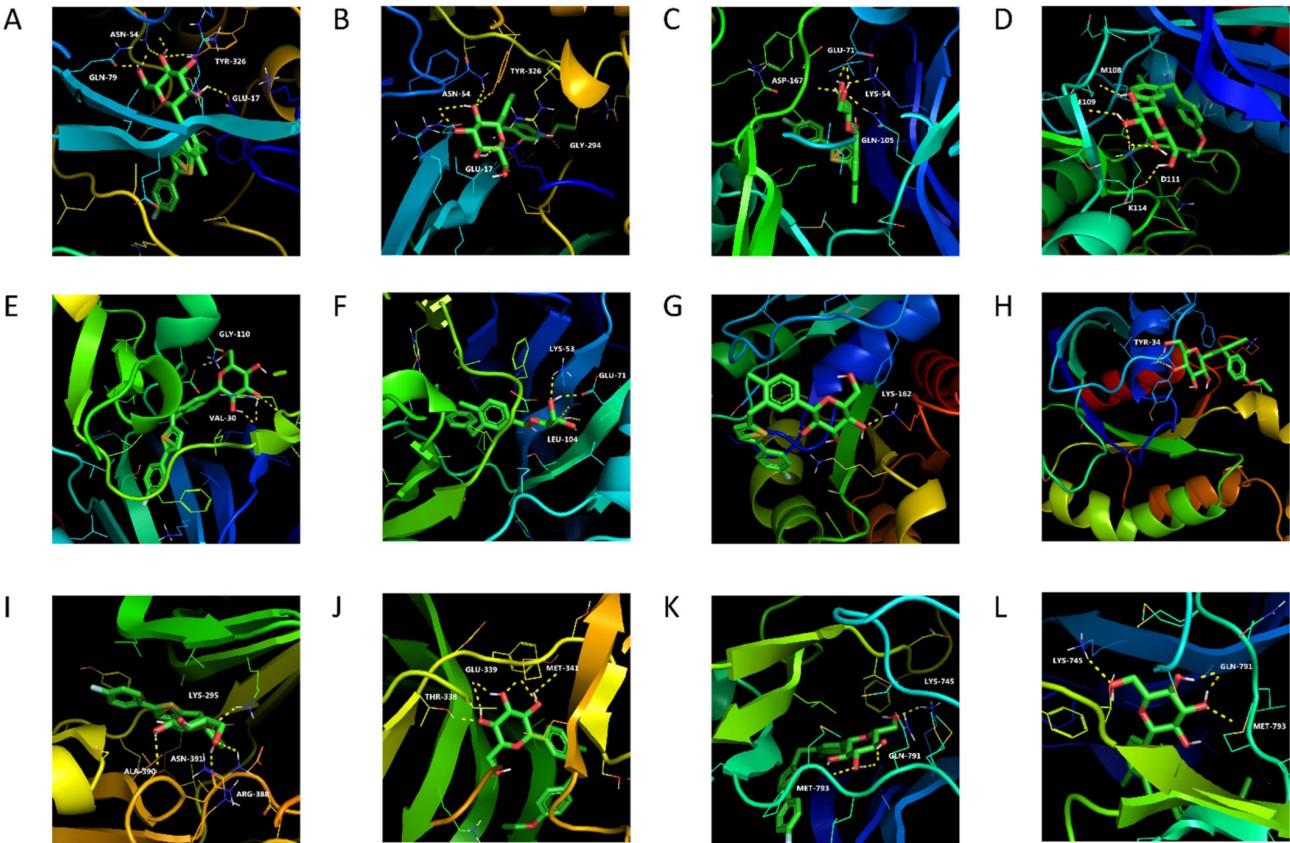

**Figure 9.** Molecular docking of canagliflozin and dapagliflozin with six overlapping targets. (**A**) Akt1-Canagliflozin; (**B**) AKT1-Dapagliflozin; (**C**) MAPK1-Canagliflozin; (**D**) MAPK1-Dapagliflozin; (**E**) MAPK14-Canagliflozin; (**F**) MAPK14-Dapagliflozin; (**G**) RHOA-Canagliflozin; (**H**) RHOA-Dapagliflozin; (**I**) SRC-Canagliflozin; (**J**) SRC-Dapagliflozin; (**K**) EGFR-Canagliflozin; (**L**) EGFR-Dapagliflozin.

**Table 5.** Molecular docking of canagliflozin with six overlapping targets.

| Gene | PDB ID | Affinity (kcal/mol) |
|:---:|:---:|:---:|
| *AKT1* | 6HHG | −9.22 |
| *EGFR* | 5UG9 | −9.81 |
| *MAPK1* | 4XJ0 | −7.79 |
| *MAPK14* | 4L8M | −10.57 |
| *RHOA* | 1A2B | −5.59 |
| *SRC* | 2H8H | −9.22 |

**Table 6.** Molecular docking of dapagliflozin with six overlapping targets.

| Gene | PDB ID | Affinity (kcal/mol) |
|:---:|:---:|:---:|
| *AKT1* | 6HHG | −7.6 |
| *EGFR* | 5UG9 | −8.31 |
| *MAPK1* | 4XJ0 | −7.31 |
| *MAPK14* | 4L8M | −11.08 |
| *RHOA* | 1A2B | −6.90 |
| *SRC* | 2H8H | −8.06 |

## 4. Discussion

In our study, we collected the relevant targets of drugs and diseases separately, took the intersection of the targets of drugs and diseases, and focused on analyzing the 191

targets that they had intersections with in the following study. The purpose of our study was to discover whether *Akt1*, *MAPK1*, *MAPK14*, *RHOA*, *SRC* and *EGFR* are central genes in different networks. These genes are both drug targets and disease-related targets. These genes, which are both drug- and disease-related targets, are six potential central targets for canagliflozin or dapagliflozin to improve AS. After verification of molecular docking, all five targets except *RHOA* were found to bind well to canagliflozin or dapagliflozin, so that *Akt1*, *MAPK1*, *MAPK14*, *SRC* and *EGFR* were determined as the final core targets.

Akt/PKB (protein kinase B) is very important for cell survival induced by growth factor. Active Akt can suppress apoptosis independently of transcription through phosphorylation and inactivation of apoptotic machine components. Akt1, Akt2 and Akt3 are three Akt isoforms in macrophages. In the mammalian genome, the major Akt isoform is encoded by Akt1, which modulates apoptosis [26]. The PI3K/Akt pathway is a classic signaling pathway that plays a crucial role in cell survival and apoptosis. Akt activated m-TOR inhibits autophagy of macrophages in the inflammatory response [27]. The change of Akt subtype or the regulation of Akt activity level significantly affects the polarized phenotype of macrophages, which may impact the progression of atherosclerosis [28]. Canagliflozin can stimulate AMPK, Akt and eNOS and inhibits iNOS and NADPH oxidase isoform 4 (NOX4), all of which are associated with antioxidant and anti-inflammatory signaling pathways [29,30]. The combined treatment with dapagliflozin and rosuvastatin can synergistically inhibit apoptosis by activating the PI3K/AKt/mTOR signaling pathway in rats with myocardial ischemia [31].

Mitogen-activated protein kinase (MAPK), a kind of serine-threonine protein kinase, plays a more important role in many important physiological and pathological processes such as cell proliferation, differentiation and apoptosis [32]. There are four main subfamilies of the MAPK pathway: ERK1/2, c-Jun N-terminal kinase (JNK), P38/MAPK and ERK5 [33]. MAPK1 (MAP kinase ERK2) is a subfamily of MAPK, form ERK1/2 [34]. MAPK14, also named as p38$\alpha$, is an isoform of the p38 MAPK family, and it is ubiquitously expressed in the family [35]. The MAPK/ERK pathway can be activated with proatherogenic stimuli in vitro and in vivo [36]. p38 MAPK is an important component of inflammatory signaling that can be activated by various stimuli such as oxidative stress, cytokines, and growth factors, all of which are involved in the formation of atherosclerosis [35,37,38].

A recent study suggests that canagliflozin has atheroprotective effects against atherosclerosis by promoting the Akt-eNOS pathway and inhibiting the activation of p38 MAPK [39]. Dapagliflozin shows a protective effect in complicated T2DM with CVD via the MAPK signaling pathway [40]. Similarly, dapagliflozin alleviates diabetic cardiomyopathy by upregulating the AKT/JAK/MAPK pathway via erythropoietin in diabetic rats [41].

Proto-oncogene tyrosine-protein kinase SRC (SRC), as other members of the SRC family kinases (SFK), plays an important role in the regulation of cellular metabolism, survival, and proliferation [42,43]. Many studies have shown that SRC plays an important role in the functional activation of macrophages and the regulation of cholesterol levels, which are involved in atherosclerosis [44,45]. A study demonstrated that canagliflozin and dapagliflozin protect endothelial cells from glucose-induced oxidative stress by blocking ROS-activated SRC, EGF receptor, protein kinase C and Rho kinase [46].

Epidermal growth factor receptor (EGFR) is a member of the ERBB family of tyrosine kinase receptors. EGFR plays a very important role in cell survival, proliferation, migration, differentiation and division [47,48]. Recently, several studies have indicated that EGFR is involved in the regulation of inflammation and oxidative stress in macrophages [49]. As we all know, inflammation and oxidative stress are significant manifestations of atherosclerosis development. Moreover, blocking EGFR induced anergia of T cells in vitro and in vivo and reduced atherosclerosis development [48]. These results suggest that EGFR plays an important role in the pathogenesis of atherosclerosis. There is currently little research on SGLT2 inhibitors and EGFR. Only one of the studies we mentioned above showed that SGLT2 inhibitors can block EGFR-related signaling pathways and play a role in protecting the vascular endothelium [46]. In our research, we found a very high degree of binding of

EGFR to both canagliflozin and dapagliflozin, so we believe that EGFR may be a key target for canagliflozin and dapagliflozin in the treatment of AS.

Arterial bifurcations and intra-arterial curves are the most common sites of atherosclerotic plaque formation [50]. The same related experimental studies have found that the endothelium of vessels in these two locations is often affected by disorder or low blood flow rate. Contrarily, vascular areas exposed to high-speed blood flow within the same vessel are less prone to form plaques [51]. Numerous experimental studies have shown that low shear stress (LSS) on the surface of vascular endothelial cells is an important factor for the occurrence and development of atherosclerosis [52]. However, there is no research on SGLT2 inhibitors to improve atherosclerosis by adjusting the shear stress. In our study, KEGG pathway analyses showed that the fluid shear stress was closely related to atherosclerosis; our research found that canagliflozin and dapagliflozin may alter fluid shear force-related pathways to improve atherosclerosis. In detail, we found that the genes *AKT1*, *MAPK14* and *SRC* are involved in fluid shear stress and the atherosclerosis pathway. In the results of the analysis of biological processes, the five core targets were mainly involved in biological processes, including cellular response to hormone stimulus, cellular response to lipid, muscle cell proliferation, a response to lipopolysaccharide, wound healing and positive regulation of cell migration. These biological processes are critical to the development of atherosclerosis. In addition, the five targets also affected molecular functions (phosphotransferase activity, protein kinase binding, protein serine/threonine kinase activity and phosphatase binding) and cellular composition (membrane raft, focal adhesion, postsynapse and postsynapse) related to atherosclerosis. Therefore, through our analysis, we speculate that canagliflozin and dapagliflozin may affect the fluid shear stress and then regulate these target genes, leading to further improvement of AS.

However, this study also has some limitations. Drug and disease targets were collected through databases with limited numbers, so some data may be biased. At the same time, since our current research results were only based on the analysis of the collected data, we need to further confirm our conclusions in future experiments.

## 5. Conclusions

Canagliflozin and dapagliflozin, potential targets, and underlying mechanisms of canagliflozin and dapagliflozin were examined using network pharmacology methods. In our study, we collected five core targets, *Akt1*, *MAPK1*, *MAPK14*, *SRC* and *EGFR,* for canagliflozin and dapagliflozin in the treatment of AS. In addition, analysis of the KEGG pathway showed that fluid shear stress and the atherosclerosis pathway were the key targets for AS treatment.

**Supplementary Materials:** The following are available online at https://www.mdpi.com/article/10.3390/jvd1010007/s1, Figure S1: The framework of the present study. Figure S2: Column chart of KEGG pathway analysis of shared targets by canagliflozin, dapagliflozin and Atherosclerosis. Figure S3: Column chart of biological process enrichment analysis of shared targets by canagliflozin, dapagliflozin and Atherosclerosis. Figure S4: Column chart of molecular functions enrichment analysis of shared targets by canagliflozin, dapagliflozin and Atherosclerosis. Figure S5: Column chart of cellular components enrichment analysis of shared targets by canagliflozin, dapagliflozin and Atherosclerosis. Table S1: Targets of canagliflozin. Table S2: Targets of dapagliflozin.

**Author Contributions:** Conceptualization, D.L.; methodology, J.W. and H.W.; software, J.W. and D.L.; validation, D.L.; formal analysis, J.W.; investigation, J.W. and W.J.; resources, W.J.; data curation, J.W.; writing—original draft preparation, W.J. and H.W.; writing—review and editing, W.J. and H.W.; visualization, W.J. and H.W.; supervision, D.L.; project administration, J.W., D.L. and H.W. All authors have read and agreed to the published version of the manuscript.

**Funding:** This research received no external funding.

**Institutional Review Board Statement:** Not applicable.

**Informed Consent Statement:** Not applicable.

**Data Availability Statement:** The data used and/or analyzed during this study are included within the article.

**Acknowledgments:** We thank Weijing Yun for supporting this study.

**Conflicts of Interest:** The authors declare no conflict of interest.

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
