# Peer review of "A Network Pharmacology to Explore the Potential Targets of Canagliflozin and Dapagliflozin in Treating Atherosclerosis"

_2813-2475, doi:10.3390/jvd1010007_

Round 1
Reviewer 1 Report
I thank you for the opportunity to comment this study.
According to the authors the main finding is following: “In our study, we collected 5 core targets Akt1, MAPK1, MAPK14, SRC and EGFR for canagliflozin and dapagliflozin in the treatment of AS.”
Comments:
- I suggest that authors use “potential targets” instead of “the mechanisms” in the title
- Abstract: Correct the following sentence “Atherosclerosis (AS) is a chronic multifactorial disease of the arterial wall and an important pathological basis of cardiovascular diseases ”
- Abstract: Correct the wording: “improving cardiovascular disease
- Introduction: Delete the first paragraph. It is not consistent. The text flows without it fluently.
- Methods: The following chapters need reference regarding methodology: 2.3; 2.5; 2.6; 2.7
- Figure 1. Need to be part of methodology. Think reader – now it is impossible to see majority of the text. Either divided in two figures or edit so that text is readable.
- Figure 2 is not readable. Edit.
- page 8 – what means that Table A3 was deleted?
- Figure 3B – not readable. Edit.
- Figure 4 (A-D) : Not readable. Edit.
- Figure 6 – use different colors – now it is impossible to see differences.
- Figure 8 – use different colors for p so that reader can see differences
- Figure 10: Use different colours for p.
- Figure 12: Please, different colours for p.
- Figure 13. It is impossible to read.
- Discussion: This type of study is always having limitations. Please, define the limitations carefully.
- Conclusion: “Our research results provide an experimental basis for further elucidating the mechanism of action of canagliflozin and dapagliflozin in the treatment of atherosclerosis.” Regarding the last statement I would be more careful because the main use of these drugs is not to prevent atherosclerosis. So, they seem to have this additional positive effect. Now the authors argue that these drugs could be used primary to prevent atherosclerosis which I think is not realistic because there are already now so many alternatives. If authors really wish to express this idea, they need to justify this carefully
- References: Be consistent how the names of the Journals are written.
Author Response
Point 1: I suggest that authors use “potential targets” instead of “the mechanisms” in the title
Response 1: Thank you for your suggestion. We have used“potential targets” instead of “the mechanisms” in the title in our new manuscript.
Point 2: Abstract: Correct the following sentence “Atherosclerosis (AS) is a chronic multifactorial disease of the arterial wall and an important pathological basis of cardiovascular diseases”
Response 2: Thanks for the valuable and meaningful suggestion. We have rewritten this sentence in our new submission.
Point 3: Abstract: Correct the wording: “improving cardiovascular disease
Response 3: Thanks for your constructive comment. We have use “the treatment of cardiovascular disease” instead of “improving cardiovascular disease” in our revised manuscript.
Point 4: Introduction: Delete the first paragraph. It is not consistent. The text flows without it fluently.
Response 4: Thank you for your important comments. We have rewritten the first paragraph in the introduction.
Point 5: Methods: The following chapters need reference regarding methodology: 2.3; 2.5; 2.6; 2.7
Response 5: Thanks for your comment. We have added relevant references in 2.3; 2.5; 2.6; 2.7 chapters.
Point 6: Figure 1. Need to be part of methodology. Think reader – now it is impossible to see majority of the text. Either divided in two figures or edit so that text is readable.
Response 6: Thanks for your comment. Figure 1 is a summary of the idea of this research. The purpose is to make more easier for readers to understand the whole content of this research. According to your comments, we have put it in the supplementary material.
Point 7: Figure 2 is not readable. Edit.
Response 7: Thanks for your valuable suggestion. We have rewritten the figure legend.
Point 8: page 8 – what means that Table A3 was deleted?
Response 8: Thanks for your meaningful comments. This is a typo and we have corrected it.
Point 9: Figure 3B – not readable. Edit.
Response 9: Thanks for your valuable suggestion. We have rewritten the figure legend.
Point 10: Figure 4 (A-D): Not readable. Edit.
Response 10: Thanks for your valuable suggestion. We have rewritten the figure legend.
Point 11: Figure 6 – use different colors – now it is impossible to see differences.
Point 12: Figure 8 – use different colors for p so that reader can see differences
Point 13: Figure 10: Use different colours for p.
Point 14: Figure 12: Please, different colours for p.
Response 11-14: Thanks for your comments. Figure 6, 8, 10 and 12 are supplementary notes to figure 5, 7, 9 and 11. The P values were given in the tables. Due to the limitation of the drawing software we used, we could not modify the color. Considering your comments and those of another reviewer, we finally decided to put these figures in the supplementary material.
Point 15: Figure 13. It is impossible to read.
Response 15: Thanks for your valuable suggestion. We have rewritten the figure legend.
Point 16: Discussion: This type of study is always having limitations. Please, define the limitations carefully.
Response 16: Thanks for your useful suggestion. We have added the limitations about this study in discussion part.
Point 17: Conclusion: “Our research results provide an experimental basis for further elucidating the mechanism of action of canagliflozin and dapagliflozin in the treatment of atherosclerosis.” Regarding the last statement I would be more careful because the main use of these drugs is not to prevent atherosclerosis. So, they seem to have this additional positive effect. Now the authors argue that these drugs could be used primary to prevent atherosclerosis which I think is not realistic because there are already now so many alternatives. If authors really wish to express this idea, they need to justify this carefully
Response 17: Thanks for your Comments. As our research still has a lot to go on, our final statement really needs to be further discussed, so we have removed this statement.
Point 18: References: Be consistent how the names of the Journals are written.
Response 18: Thanks for your valuable suggestion. We have modified the format of the references according to the requirements of the journal.
Reviewer 2 Report
The manuscript entitled "A Network Pharmacology to Explore the Mechanism of Canagliflozin and Dapagliflozin in treating Atherosclerosis" presents a series of analyses to pinpoint target genes for Canagliflozin and Dapagliflozin. I have many concerns I would like to address with the authors:
1. Atherosclerosis is a very complex molecular process, with several inflammatory mechanisms. Diabetes also has an inflammatory profile. These processes are not presented in the text or the analysis, and I have missed the link between atherosclerosis, hypertension, and other vascular disorders.
2. In the Introduction, the authors mention: "In clinical trials, in T2DM patients with atherosclerotic cardiovascular disease or risk factors, treatment with dapagliflozin results in a lower rate of cardiovascular death or hospitalization for heart failure (11)".
Reference 11 (Wiviott et al., 2019) says: "The lower rate of cardiovascular death or hospitalization for heart failure in the dapagliflozin group than in the placebo group was consistent across multiple subgroups, which shows that dapagliflozin prevented cardiovascular events, particularly hospitalization for heart failure, in a broad range of patients, regardless of a history of atherosclerotic cardiovascular disease or heart failure.
This reference does not justify the effect of dapagliflozin on atherosclerosis. As I mentioned, vascular disease mechanisms are complex, and many molecular studies must be performed before assuming this mechanism exists.
3. The authors have performed the GO and KEGG pathways enrichment analyses for the 191 genes in common between "atherosclerosis", "canagliflozin", and "dapagliflozin". Of course, this strategy would lead to several genes related to "atherosclerosis". The authors should not have included atherosclerosis in this analysis and evaluated the enrichments of the 288 canagliflozin targets and 287 dapagliflozin targets instead.
4. AKT1, EGFR, and MAPK1 are central genes in different networks, because of their role as upstream regulators of several pathways. The analysis here performed is not experimental or specific to the point to confirm the drugs act on these genes, especially by treating atherosclerosis.
5. There are too many images without proper figure legends explaining their results. Many figures are not essential for the manuscript and should be moved to Supplementary Material.
6. Minor point: genes should be italicized.
Author Response
Point 1: Atherosclerosis is a very complex molecular process, with several inflammatory mechanisms. Diabetes also has an inflammatory profile. These processes are not presented in the text or the analysis, and I have missed the link between atherosclerosis, hypertension, and other vascular disorders.
Response 1: We appreciate the reviewer’s valuable suggestion. We have re-write the introduction.
Point 2: In the Introduction, the authors mention: "In clinical trials, in T2DM patients with atherosclerotic cardiovascular disease or risk factors, treatment with dapagliflozin results in a lower rate of cardiovascular death or hospitalization for heart failure (11)".
Reference 11 (Wiviott et al., 2019) says: "The lower rate of cardiovascular death or hospitalization for heart failure in the dapagliflozin group than in the placebo group was consistent across multiple subgroups, which shows that dapagliflozin prevented cardiovascular events, particularly hospitalization for heart failure, in a broad range of patients, regardless of a history of atherosclerotic cardiovascular disease or heart failure.
This reference does not justify the effect of dapagliflozin on atherosclerosis. As I mentioned, vascular disease mechanisms are complex, and many molecular studies must be performed before assuming this mechanism exists.
Response 2: Thanks for your comments. What we want to express here is that dapagliflozin can be used to treat cardiovascular disease, which is not accurate enough. We agree with your statement that vascular disease mechanisms are complex, so we hope through our research found some new targets, and we will verify the specific molecular mechanism in the next study. We have revised this statement in the latest manuscript based on your suggestion.
Point 3: The authors have performed the GO and KEGG pathways enrichment analyses for the 191 genes in common between "atherosclerosis", "canagliflozin", and "dapagliflozin". Of course, this strategy would lead to several genes related to "atherosclerosis". The authors should not have included atherosclerosis in this analysis and evaluated the enrichments of the 288 canagliflozin targets and 287 dapagliflozin targets instead.
Response 3: Thanks for your valuable suggestion. In our study, we collected the relevant targets of drugs and diseases separately, and took the intersection of the targets of drugs and diseases, and focused on analyzing the 191 targets that they have intersection in the following study The purpose of doing this is to discover the core targets and related metabolic pathways of canagliflozin and dapagliflozin for the treatment of atherosclerosis. If only the targets of the drugs are analyzed, the targets that may be found are not related to the disease.
Point 4: AKT1, EGFR, and MAPK1 are central genes in different networks, because of their role as upstream regulators of several pathways. The analysis here performed is not experimental or specific to the point to confirm the drugs act on these genes, especially by treating atherosclerosis.
Response 4: Thanks for your useful suggestion. The purpose of our study is to discover that like AKT1, EGFR, and MAPK1 are central genes in different networks. These genes are both drug targets and disease-related targets, but as you said, whether canagliflozin and dapagliflozin can treat atherosclerosis through these genes needs further experimental verification, which is the focus of our next research.
Point 5: There are too many images without proper figure legends explaining their results. Many figures are not essential for the manuscript and should be moved to Supplementary Material.
Response 5: Thanks for your constructive comment. We have moved some figures to Supplementary Material.
Point 6: Minor point: genes should be italicized.
Response 6: Thanks for your valuable suggestion. We have corrected.
Round 2
Reviewer 1 Report
Dear authors,
Thank you for comments.
Figure 3 A & B are still unreadable. Can you make them bigger or modify otherwise?
Author Response
Point 1: Figure 3 A & B are still unreadable. Can you make them bigger or modify otherwise?
Response 1: Thank you for your important comments. We have made a more detailed explanation of the Figure 3 A & B.
Reviewer 2 Report
The authors have answered my concerns, however, the responses regarding the central genes in the network and the strategy to use the intersection with atherosclerosis must be better explained in the text, not only in the author's reply.
Author Response
Point 1: The authors have answered my concerns, however, the responses regarding the central genes in the network and the strategy to use the intersection with atherosclerosis must be better explained in the text, not only in the author's reply.
Response 1: Thanks for the valuable and meaningful suggestion. We have provided a better explanation of the core targets and the strategies we used in the text.